# Research on Residents' Travel Behavior under Sudden Fire Disaster Based on Prospect Theory

**Ciyun Lin** [1,2], **Kang Wang** [1,2], **Dayong Wu** [3] **and Bowen Gong** [1,2,*]

1   Department of Traffic Information and Control Engineering, Jilin University, Changchun 130022, China;
    linciyun@jlu.edu.cn (C.L.); kangwang18@mails.jlu.edu.cn (K.W.)
2   Jilin Engineering Research Center for ITS, Changchun 130022, China
3   Texas A&M Transportation Institute, Texas A&M University, College Station, TX 77843, USA;
    J-Wu@tti.tamu.edu
*   Correspondence: gongbowen@jlu.edu.cn

**Abstract:** The decision-making process of travel behaviors under uncertainty and risk shall be analyzed in order to solve the emergency traffic management or evacuation problem under sudden fire disaster in a high-density urban environment. Firstly, this paper attempts to acquire the travel risk attitude thought online survey questionnaires. In the questionnaire, we focused on obtaining the traveler's response thought set a scene and obtain the traveler's risk attitude. Secondly, we explore the relationship between traveler's personal attributes and risk attitudes through questionnaires. Finally, the questionnaire data were used to calibrate and adjust the parameters in the proposed prospect theory (PT) based model. Subsequently, the K-T model and Wang's model were used to compare and verify the accuracy and validity of the proposed model. The results presented that the proposed model is more accurate and the largest prediction error of travel route selection behavior is only nine percent.

**Keywords:** sudden fire disaster; prospect theory; path choice; risk attitude; SP questionnaire

## 1. Introduction

Urban population, buildings, and other types of infrastructure are more highly concentrated with the rapid development of China's community economy and the accelerating urbanization process, which greatly increases the probability of sudden fire incidents in high-density urban environment, especially in the urban center. The impact scope of sudden fire disaster in high-density urban environment is getting wider and wider, and the property losses are mounting. Fast and effective emergency traffic organizational guarantee can significantly reduce loss of life and property. Deep analysis and understanding travel behavior and characteristic of travel selection is the first step in making an effective emergency traffic management and control in sudden fire disaster. Accordingly, it is necessary to study the impact of sudden fire disaster on travelers in travel decision-making.

When travelers choose the road, they will be affected by external factors and their own attributes. Many scholars combine the two factors to analyze their travel behavior, trying to provide better strategies for traffic management. Carlo G P, Shlomo B, and Cristina P [1] consider potential variables (such as memory, habit, familiarity, spatial ability, time-saving skills) together with traditional variables (such as travel time, distance, and congestion degree), which enrich the understanding of route selection behavior. Xuan Di [2] introduced static traffic assignment and dynamic traffic assignment into entity bounded rationality and analyzed the prediction of travel behavior that is based on bounded rationality. Heilman, R. M. [3] found that emotion regulation is very important to the decision-making rationality

of drivers, especially in risky and uncertain environments. These studies have shown that different attributes of their own have an impact on the choice of roads.

At the same time, many scholars have undertaken research on the choice of travel path in the uncertain environment. They used models include prospect theory, regret theory, psychological account theory, and so on. Based on the study of traveler's travel risk assessment, Tian [4] found that travelers' psychological factors have greater impact on traveler's route choice. Ramos [5], while using experimental survey data to compare the expected utility theory, prospect theory and regret theory in the travel path selection prediction and found that the prospect theory of behavior is more applicable than the theory of expected utility in the travel path selection. Michea [6] and Bhat [7] concluded that prospect theory is more suitable for the path selection behavior of travelers than expected utility theory in uncertain environments through case studies. Zhao Wei and Zhang Xing-chen [8], from the establishment of a single travel theory selection model, when compared the travel path selection model based on prospect theory and based on the expected utility theory, indicated that the prospect theory could overcome the shortcomings of expected utility to some extent. Jou Rongchang [9] uses prospect theory to reflect the risk attitude of Taiwan Highway drivers, and concludes that most drivers' driving behaviors conform to the characteristics of prospect theory. From the current research status quo, prospect theory has achieved good results in the study of traveler path selection and it is more suitable for describing the path choice of travelers.

The importance of uncertainty in travel decision-making process has attracted increasing researchers' attention. With uncertainty as the theoretical framework, researchers have carried out a lot of research on the impact of individual psychological characteristics, traffic system operation, intelligent information system, and other factors on travelers' route choice behavior in travel decision-making process. In the research work, the prospect theory is deeply analyzed in the traffic travel behavior. Prospect theory combines economics and psychology, and it describes the utility of travelers' choice scheme through value function and weight function. Value function and weight function together determine the attitude of decision makers to uncertain events and influence the behavior decision of travelers. Kahneman and Tversky [10] have given the values of relevant parameters in the prospect theoretical model on the basis of economic experiments. Wu [11], Prelec [12], and Senbil [13] in the empirical research showed that the relevant parameters in the prospect theory were not fixed, and their values depend on the decision-making environment, and different decision makers would also have different results. Many scholars have also considered the influence of individual attributes, such as gender and age, on relevant parameters in prospect theory. For example, Fehr-Duda [14] studied the relationship between gender and risk taking, and the results showed that women were more risk averse than men in general. Wang Kai [15] conducted a study on the travel route choices of travellers in Chao yang District, Beijing, and found that the risk appetite of travellers in different decision scenarios was significantly different. Avineri and Bovy [16] pointed out that the change of relevant parameters in the prospect theory affects the prediction result when using the prospect theory to analyze the path selection behavior of the traveler. When using the prospect theory to analyze the travel selection behavior, it should be based on the travelers' risk attitudes to assign appropriate values to relevant parameters. Wang Yan [17] used economics to classify travelers according to risk attitudes. Through questionnaires, travelers with different risk attitudes assigned different parameters that were more in line with actual behavior. Neilson [18] found, in the research, that the relevant parameters in the prospect theory cannot be applied to different fields, and the research in different fields should assign corresponding parameters to different problems.

The difference in travel modes and the different attributes of travelers under sudden fire disaster pose a challenge for emergency evacuation and emergency traffic management. Sudden fires have uncertainties in location and size when compared with other natural disasters, making the choice of escape route more unique and different from the choice of escape route under other disasters. It is different from the study of road networks around disasters in the past sudden disasters. Travelers and

surrounding road networks are used to analyze the path selection behavior of specific travelers under sudden fire disaster. This paper mainly focuses on:

(1) Applying the prospect theory of investigating investor risk attitudes in economics to the risk attitude of travelers under sudden fire disasters, because there is commonality in analyzing people's risk choices.

(2) In the study of emergency traffic in sudden fire disaster areas, this paper takes travelers as the research object and, through questionnaire survey, start from the physiological psychology of the travelers, and set a risk attitude discrimination scene to class travelers according to the type of risk attitude, and the analysis of the traveler in this paper is more detailed.

(3) In the past studies, usually, only a single mode of travel was studied. In this paper, we take the fire in the central area of a city as an example, and divide the traveler into driving and walking, which basically covers all of the travelers in the fire area and different travel modes. The travelers who different travel modes are classify to study, and each type of travel mode sets up six scenarios, which makes the selection result more reliable, and obtains corresponding evacuation measures to solve the congestion problem around the fire more quickly.

The remainder of this paper is organized, as follows. Section 2 is a detailed introduction to the theoretical part of the prospect theory. Section 3 investigates the personal attributes of travelers through a questionnaire survey and sets up a scene to classify the risk attitudes of travelers. In Section 4, the model analysis of the traveler's path selection is made. Subsequently, a sudden fire disaster in the central area of city is used as an example and set up different travel scenarios. According to the path selection result of the respondents, the value function of the prospect theory is calibrated, and the parameter results are obtained. Finally, the results of this paper are compared with the calibration results of K-T(Kahneman and Tversky) [10] model and Wang Yan's model [17]. Section 5 introduces the research conclusions and future research prospects.

## 2. Methods

Prospect theory mainly reflects people's sensitivity to change. "Benefit" and "loss" are used to measure people's decision-making and can be used to study the risk attitude of travelers in decision-making. In this paper, we used prospect theory to analyze the decision-making process of decision makers under uncertain conditions and divided the decision-making process into two stages: editing and evaluation. The editing phase compares the possible outcomes of the alternatives with reference points, describes each alternative, and then simplifies the editing of the alternatives to facilitate decision makers to make choices. After the editing stage, the decision makers began to evaluate the various options, while using the value function and the decision weight function to calculate the prospect value of the alternative, and make a choice by comparing the size of the prospect value of the alternative.

### 2.1. Value Function

The value function represents the expected result of the individual's evaluation of the selection scheme $x$. The traveler's perception of the scheme $x$ depends on the reference point of subjective perception. The reference point is used to judge the gap between the scheme $x$ and the standard. When $x \geq 0$ indicates benefit, $x < 0$ indicates loss. The traveler is more sensitive to the loss when compared with the benefit. There is asymmetry between the feeling and the loss feeling. The perception of the scheme can be described by the value function $V(x)$:

$$V(\Delta x) = \begin{cases} \Delta x^{\alpha} & \Delta x \geq 0 \\ -\lambda(-\Delta x)^{\beta} & \Delta x < 0 \end{cases} \tag{1}$$

where $\Delta x = x - x_0$ is the deviation of the possible result $x$ of the scheme from the reference points. $\alpha$ and $\beta$ are the risk preference coefficient of the decision maker, $0 < \alpha < 1, 0 < \beta < 1$. $\lambda$ is the loss

avoidance coefficient and $\lambda \geq 1$. The larger the value, the greater the loss avoidance of the decision maker. Abdellaoui [19] research results showed that, when $\alpha = 0.725$, $\beta = 0.717$, and $\lambda = 2.04$, the value function changes in the yield area and the damaged area, as shown in Figure 1:

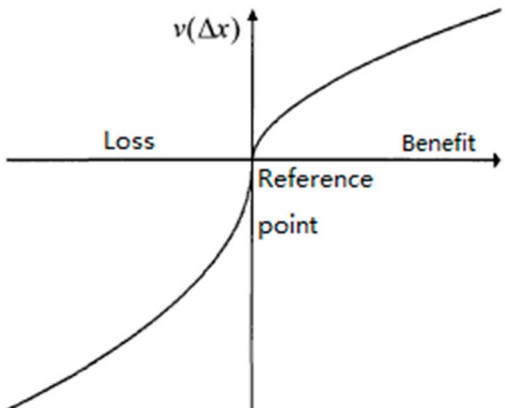

**Figure 1.** Value function diagram.

The prospect theoretical value function has the following characteristics:

(1) The decision makers mainly focus on the deviation from the reference point in the decision-making, and the value function in the prospect theory is divided into two areas of income and loss by the reference point.

(2) The value function is S-shaped. The decision makers tend to avoid risk in the income area. In the loss area, the decision makers tend to pursue risk, and the curve in the loss area is steeper, which indicated that the decision makers are more sensitive to the change of losses and the change of income. The slope is less than the slope of the loss change.

*2.2. Weight Function*

The decision weight function represents a subjective assessment of objective probability, $W(P)$ reflects the probability assessment of the decision makers on the prospective outcome, which reflects the decision-maker's perception of the probability of occurrence of the risk event, and its parameters indicate the sensitivity of the decision maker to the objective probability change. $p$ indicates the probability of occurrence of $x$; $W(P)^+$, $W(P)^-$, respectively, represent the decision weights defined in the income and loss areas. The decision weight function is an inverted "S" shape. The smaller the parameters $\gamma$ and $\delta$ ($0 < \gamma < 1$, $0 < \delta < 1$), the more curved of the function is. The specific expressions are shown in formula (2) and formula (3). The parameter values were used in this paper, as in Kahneman and Tversky [10]. $\gamma = 0.61$, $\delta = 0.69$.

$$W(P)^+ = \frac{P^\gamma}{\left(P^\gamma + (1-P)^\gamma\right)^{-\gamma}} \tag{2}$$

$$W(P)^- = \frac{P^\delta}{\left(P^\delta + (1-P)^\delta\right)^{-\delta}} \tag{3}$$

The weight function changes in the yield area and the damaged area, as shown in Figure 2:

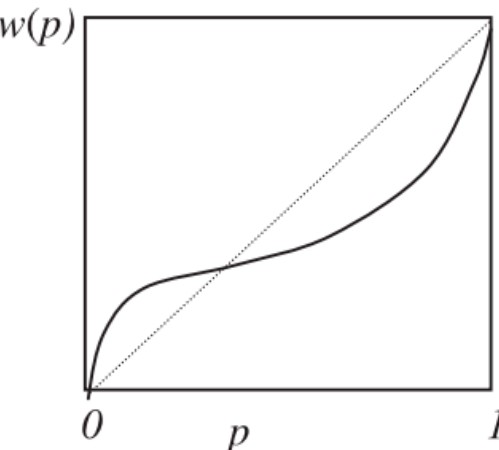

**Figure 2.** Weight function diagram.

*2.3. Prospect Value Function*

According to the cumulative prospect theory, the prospect value of alternative $f$ is as in formula (4):

$$V(f) = \sum_{i=1}^{n} \pi^+(p_i)v^+(x_i) + \sum_{j=-1}^{-m} \pi^-(p_j)v^-(x_j) \qquad (4)$$

## 3. Survey and Analysis

The SP survey is widely used in traffic surveys [20]. In this paper, we first set up a travel scenario to classify the risk attitudes of travelers. Travelers are classified into three categoriesA according to different choices: risk pursuit, risk neutrality, and risk aversion, so that travelers with different risk attitudes can be classified and studied under sudden fire. Secondly, we investigate the personal attributes of travelers and find out the relationship between personal attributes and risk attitudes, so as to locate the types of risk attitudes of travelers more accurately.

We conduct online questionnaire survey through the most authoritative "questionnaire star" application in China, which can be spread in the form of Wechat group and QQ group, and the application can count the final survey results. We conduct communication survey through Wechat group and circles of friends, and there is no time limit to ensure that the respondents have sufficient time to make serious choices. According to the types of travelers in the city, the Wechat group spread includes the group of ordinary office workers, the group of government employees, the group of teachers and college students, the group of the elderly, etc., which basically includes all of the travel groups in the city. We conducted a three-month investigation from 1 November 2018 to 31 January 2019. From the beginning of the survey, we have made a comprehensive consideration of the respondents to prevent the occurrence of unreasonable data as much as possible, and from the statistical results after processing, the survey data that were obtained are in line with our expectations, and we think that are basically in line with the actual travel situation.

In this paper, uncertainty and risk represent different meanings. In the investigation of risk attitude, we assume that there are uncertain factors in the road conditions, and the risk is that travelers may be late when choosing different roads, and they will be fined when they are late. Therefore, it is different between the uncertainty factors of the road and the risk of fines that travelers may have. In the investigation of the travel behavior in case of sudden fire, the time of the road, and arrival probability in each scene, we design it according to the prospect theory. Assuming that the conditions of all the travel roads are basically similar, in each scenario, the road that is affected by the fire might have the risk of congestion or life-threatening, and there will also be uncertainty factors for smoother traffic. Congestion is due to the chaos of traffic order at the scene of fire. Smoothness is due to the police at the

scene of fire have made timely traffic control, which is smoother than in normal conditions, and these uncertain factors of benefit or damage just accord with the characteristics of prospect theory. In our questionnaire, we investigated the drivers of traffic characteristics, while ignoring the drivers who traveled through them.

## 3.1. Risk Attitude Survey

In this paper, the risk attitude of travelers in route selection is acquired through network questionnaire. Specific situations are set up in the questionnaire, and the acceptable probability of travelers is asked to obtain the risk attitude. Figure 3 shows scene settings in the survey:

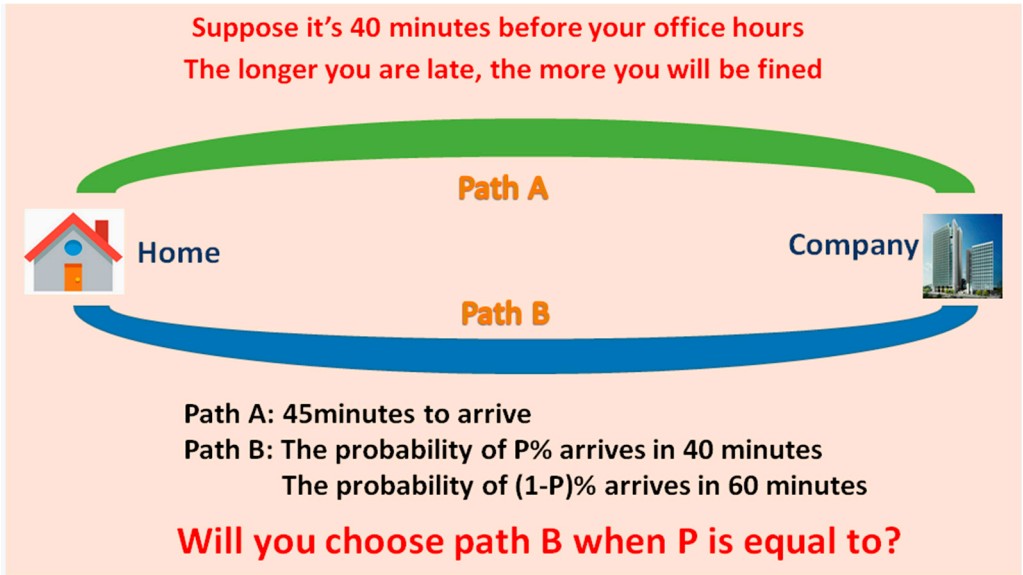

**Figure 3.** Travel attitude analysis scene graph.

The options are: A: 0–30%; B: 30–60%; and, C: 60–90%.

In the above scenario settings, path A is not risky and sure to be late; path B might be late or might be on time. The lower the probability of choosing to arrive in time, the more the traveler tends to pursue risk and the opposite is risk aversion. In economics, risk attitudes are generally divided into three categories [21]: risk pursuit, risk neutrality, and risk aversion. According to the risk attitude, the travelers are divided into three categories, 0–30% is classified as risk pursuers, 30–60% is classified as risk neutrals, and 60–90% is classified as risk averse.

In this survey, 349 questionnaires were collected through an online survey, and the incomplete and unreasonable questionnaires were removed. Finally, 310 valid questionnaires were obtained, with an effective rate of 88.9%.

Figure 4 shows the risk attitude results, according to the survey data, in which 120 people are risk-seekers, accounting for 38.7%. 100 people with risk neutrality, accounting for 32.3%. 90 people with risk aversion, accounting for 29%.

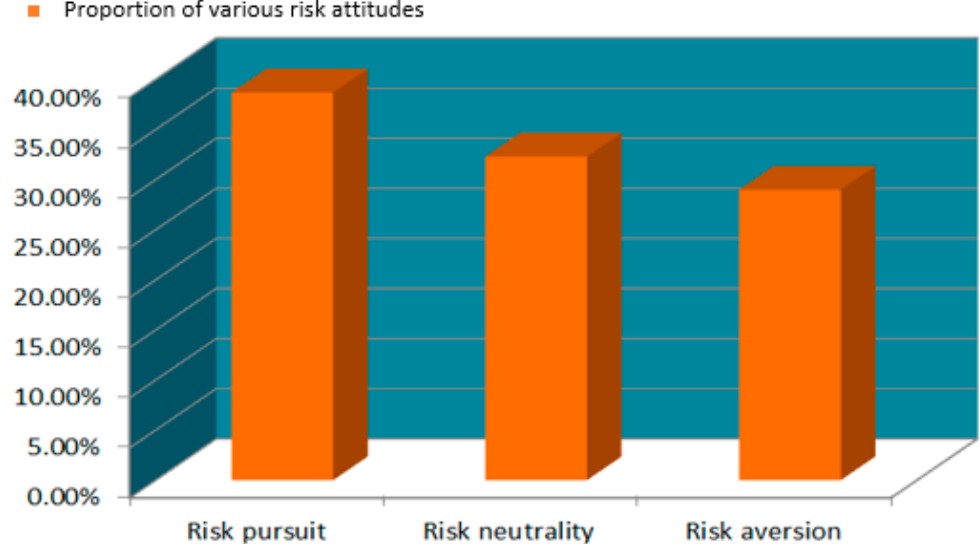

**Figure 4.** Analysis of the risk attitude of the traveler.

### *3.2. Traveler Personal Attributes*

In the survey of personal attributes of travelers, personal attribute questionnaire is shown in Table 1. Table 2 shows the final results of the questionnaire:

**Table 1.** Personal attribute questionnaire.

| | | | |
|---|---|---|---|
| **1. Your Gender is ( )** | | | |
| A. Male | | B. Female | |
| **2. Your Age is ( )** | | | |
| A. 18–30 | B. 31–40 | C. 41–50 | D. 51–60 |
| **3. Your Career is ( )** | | | |
| A. Civil servants and institutions | B. Corporate staff | C. Self-employed households | |
| D. Student | E. Other | | |
| **4. Your Driving Age is ( )** | | | |
| A. Less than one year | | B. One to three years | |
| C. Three to five years | | D. More than five years | |
| **5. Your Monthly Income (RMB) is ( )** | | | |
| A. Less than 2000 | B. 2000–5000 | C. More than 5000 | |
| **6. Your Education Background is ( )** | | | |
| A. High school and below | | B. Undergraduate | |
| C. Master degree and above | | D. Master degree and above | |

In the questionnaire, we only consider six indicators as influencing factors. In addition to the six basic attributes, other factors, such as attitude and perception, will also have an impact on travel behavior. Because we are investigating the basic attributes of all travelers in the city, the impact of attitude and perception needs to be analyzed according to different environments. Hence, we only take personal attributes as influencing travel behavior factors in this survey.

**Table 2.** Questionnaire results.

| Personal Attribute | | Survey Results | Personal Attribute | | Survey Results |
|---|---|---|---|---|---|
| Gender | Male | 50.3% | Driving age | Less than one year | 43.5% |
| | Female | 49.7% | | One to three years | 14.8% |
| Age | 18–30 | 23% | | Three to five years | 13.6% |
| | 31–40 | 32% | | More than five years | 28.1% |
| | 41–50 | 31% | Monthly income (RMB) | 2000 and below | 21.9% |
| | 51–60 | 14% | | 2000–5000 | 59.7% |
| Career | Civil servants and institutions | 14.8% | | More than 5000 | 18.4% |
| | Corporate staff | 41.6% | Education background | High school and below | 10.9% |
| | Self-employed households | 11% | | Undergraduate | 53.2% |
| | Student | 19.4% | | Master degree and above | 27.7% |
| | Other | 13.2% | | Other | 8.2% |

From the Table 2, we can find that: in the questionnaire survey, the number of men and women is basically the same; the subjects of the survey are 18–60 years old travelers, mainly between 18–50 years old, accounting for 86%; among the respondents, 75.8% of the regular commuters, who have a high demand for time punctuality; and 56.5% of the people have more than one year driving experience. The monthly income of the sample is mainly concentrated in 2000–5000 RMB, accounting for 59.7%, and most of the travelers have higher education. Generally speaking, the survey results are in line with the characteristics of most travelers in cities.

*3.3. Optimal Scale Regression Analysis*

In a sudden disaster environment, travelers will face different decision-making choices under the influence of psychophysiology. Individual attributes affect the risk attitudes of travelers. The individual attributes of travelers are used as explanatory variables for explaining risk attitudes, and the influencing factors of risk attitudes are analyzed by using survey data.

When analyzing the factors affecting the risk attitude of the traveler, the analysis is performed while using the optimal scale regression model, as shown in formula 5:

$$Y = b_1 X_1 + b_2 X_2 + b_3 X_3 + \ldots + c \tag{5}$$

where $Y$ represents the variable to be interpreted. This paper shows the type of risk attitude of the traveler in a sudden fire disaster. Given the corresponding value, risk pursuit = 1, risk neutral = 2, risk aversion = 3. $X_i$ is the explanatory variable, which points out the personal attributes of the walker in this article. $b_i$ is the undetermined coefficient. $c$ is random perturbation and $E(c) = 0$, $\text{var}(c) = \delta^2$. Table 3 shows the explanation of each variable in the model:

**Table 3.** Explanation of model variables.

| Name | Symbol | Variables and Assignments | |
|---|---|---|---|
| Gender | $X_1$ | Male | 1 |
| | | Female | 2 |
| Age | $X_2$ | 18–30 | 1 |
| | | 31–40 | 2 |
| | | 41–50 | 3 |
| | | 51–60 | 4 |
| Career | $X_3$ | Civil servants and institutions | 1 |
| | | Corporate staff | 2 |
| | | Self-employed households | 3 |
| | | Student | 4 |
| | | Other | 5 |
| Education background | $X_4$ | High school and below | 1 |
| | | Undergraduate | 2 |
| | | Master degree and above | 3 |
| | | Other | 4 |
| Monthly income (RMB) | $X_5$ | Less than 2000 | 1 |
| | | 2000–5000 | 2 |
| | | More than 5000 | 3 |
| Driving age | $X_6$ | Less than one year | 1 |
| | | One to three years | 2 |
| | | Three to five years | 3 |
| | | More than five years | 4 |

### 3.4. Analysis of the Affecting Factors

The effective questionnaire was 310, and six explanatory variables were selected. In the regression analysis experience, the regression analysis was considered to be effective when the sample reached 30 times of the calibration coefficient. Therefore, this survey can meet the requirements of regression analysis.

Regression analysis of the data while using IBM SPSS 22.0 software ((IBM, New York, NY, USA)), the relationship between travel decision and individual attributes of the traveler is obtained. The calibration results are shown in Table 4. The coefficient analysis is analyzed in Tables 5 and 6 shows the correlation analysis of the influencing factors:

**Table 4.** Model coefficient determination.

| Complex Correlation Coefficient R | Decision Coefficient $R^2$ | Correction Coefficient $R^2$ | F | *Sig* |
|---|---|---|---|---|
| 0.218 | 0.215 | 0.167 | 2.543 | 0.003 |

As shown in Table 4, as above, the judgment coefficient of the model is 0.215, and the correction coefficient is 0.167, which indicates that the model can fit the data to a certain extent, and the coefficient and correction coefficient are small, indicating that other information that affects the risk attitudes of travelers is not included in this paper and further research is needed in the future. The significance of the model t-test was 0.003, which was less than 0.05, which indicated that the regression model was statistically significant.

**Table 5.** Coefficient analysis.

| Variable | Standard Coefficient | | df | Sig |
|---|---|---|---|---|
| | a | Standard Error | | |
| Gender ($X_1$) | 0.540 | 0.079 | 1 | 7.943 | 0.001 |
| Age ($X_2$) | 0.147 | 0.045 | 1 | 0.983 | 0.001 |
| Career ($X_3$) | 0.025 | 0.030 | 4 | 2.943 | 0.398 |
| Education background ($X_4$) | −0.005 | 0.039 | 1 | 1.762 | 0.899 |
| Monthly income ($X_5$) | −0.068 | 0.062 | 2 | 1.48 | 0.275 |
| Driving age ($X_6$) | −0.113 | 0.040 | 3 | 4.431 | 0.004 |

Formula (6) shows the fitted regression equation that can be obtained from the above table:

$$y = 0.54x_1 + 0.147x_2 + 0.025x_3 - 0.005x_4 - 0.068x_5 - 0.113x_6 \qquad (6)$$

**Table 6.** Analysis of the correlation of influencing factors.

| Variable | Correlation | | Importance | | Tolerance | |
|---|---|---|---|---|---|---|
| | 0 th Order Correlation | Partial Correlation | Partially Related | | After Transformation | Before Transformation |
| Gender ($X_1$) | 0.393 | 0.375 | 0.354 | 0.615 | 0.958 | 0.948 |
| Age ($X_2$) | 0.140 | 0.187 | 0.167 | 0.106 | 0.916 | 0.903 |
| Career ($X_3$) | 0.108 | 0.106 | 0.093 | 0.144 | 0.978 | 0.962 |
| Education background ($X_4$) | −0.068 | −0.036 | -0.032 | 0.110 | 0.934 | 0.955 |
| Monthly income ($X_5$) | −0.151 | −0.060 | -0.053 | 0.038 | 0.824 | 0.680 |
| Driving age ($X_6$) | −0.242 | −0.181 | -0.161 | 0.188 | 0.807 | 0.685 |

Table 6 gives the analysis, importance, and tolerance of each explanatory variable for risk attitude correlation. It can be seen from the table that the importance of monthly income is less than 0.1, which indicates that the monthly income has a low correlation with the risk attitude of the traveler, and the other five variables have a strong collinear relationship with the risk attitude of the traveler.

From the regression equations fitted above, the following conclusions can be drawn:

(1) Gender is positively correlated with the risk attitude of travelers, which indicated that, under uncertainty, men are more prone to risk than women, consistent with the risk study of decision makers in economic investment.

(2) The age is positively related to the risk attitude of the traveler. It is the same as the economic investment and it might be related to the rich travel experience of the older.

(3) The education level is positively correlated with the risk attitude of the traveler, that is, the higher the education level, the more the traveler is biased towards risk avoidance, which is consistent with the research results of Xu Hongli [22,23].

(4) The occupational type is negatively correlated with the risk attitude of the traveler. The employees of the enterprise and the business unit are more inclined to pursue the risk under the uncertainty factor, while the students prefer risk avoidance, which indirectly indicates that the education level is higher. Under the disaster, it is biased towards risk aversion.

(5) The driving age is negatively correlated with the risk attitude of the traveler, which indicates that, the longer the driver is, the more experienced the traveler is, and the more likely the risk is to avoid the sudden risk.

(6) The monthly income is negatively correlated with the risk attitude of the traveler, that is, the higher the income of the traveler is more inclined to risk pursuit, which is consistent with the research conclusions in economic investment.

Through the analysis of the six factors mentioned above, it shows that there is a great relationship between the types of travelers' risk attitudes and their personal attributes. We studied the different

types of people with different risk attitudes under sudden fire, according to the different types of risk attitudes.

## 4. Path Selection Based on Prospect Theory

The traffic conditions of the road and the characteristics of the traveler both influence the choice of traveler's travel path. Different travelers will have different risk attitudes under sudden fire disaster and will have different performances in path selection. Based on the prospect theory, this paper establishes a travel route selection model for travelers under sudden fire disaster and then analyzes the travelers in different risk attitude types. The third section has divided the travelers into three categories, in this section, setting up the travel scene of sudden fire, and studying the choice behavior of the three risk attitude groups of people.

### 4.1. Travel Route Selection Model

The prospect-based travel route selection model under sudden fire disaster is established while combining the prospect theory and the traveler route selection block diagram introduced in Section 2. In this study, the passage time of the whole journey will be taken as a reference point.

Suppose that the traveler R is on a travel:

$(X^j, P^j)$ is travel time of path $R^j$, $j$ is a possible alternative path in travel, $j = 1, 2, \ldots n$, $R^j$ possible time is $X^j = \{x_1^j, \ldots x_n^j\}$, and the probability of its occurrence is $P^j = \{p_1^j, \ldots p_n^j\}$, $p_1^j + p_2^j + \ldots p_n^j = 1$, Travel time reference point is $x_0$. Subsequently, the prospect value $V^j$ of $R^j$ is shown in formula 7:

$$V^j(x_1^j, p_1^j; x_2^j, p_2^j; \ldots, x_n^j, p_n^j) = \sum_{i=1}^{n} v^j(\Delta x_i^j)\pi^j(p_i^j) \tag{7}$$

where $v^j(\Delta x_i^j)$ is the value function of path $R^j$ and $\pi^j(p_i^j)$ is the cumulative weight function of path j. The traveler selects the probability $P^j$ of the path $R^j$ and uses the Logit selection model to calculate the formula as shown in formula (8):

$$P^j = \frac{\exp(V^j)}{\sum_{j=1}^{j} \exp(V^j)} \tag{8}$$

where $R^j$ is traveler R on the path $j$, $P^j$ is the probability that the traveler chooses path $j$, A represents the prospect value of path j, and the traveler finally selects the path with the largest prospect value, and the prospect value of each path is calculated by the prospect value formula.

The experimental design follow the methods of Xu [24], Shi [25], Yang J [26], and it is assumed in the questionnaire that a fire happened in the center of a city. In the questionnaire, according to prospect theory, the path selection of residents in the "benefit" and "loss" scenarios under sudden fire disaster is set. There are drivers and walkers traveling from the designated starting point to the designated destination point. Additionally, there are only two paths to be chosen, path A and path B. The path travel time is also uncertain due to the uncertainty of the size of the sudden fire disaster. Additionally, the path through the fire area is more time volatile. The scenario scenes were set, as follows:

Scenario 1: Assuming that the traveler has two paths to choose from the designated starting point to the designated destination (Path A and Path B, one of which passes through the fire area), the travel time of the two paths is the same. The travel time of the two paths is reduced because of traffic control in the fire surrounding area. We believe that travellers benefit in this case. The travel time of path A is $(t_1, t_2, t_3)$, the corresponding probability is $(p, q, 1 - p - q)$, the travel time of path B is $(t_4, t_5)$, and the corresponding probability is $(r, 1 - r)$.

Scenario 2: Assuming that the traveler has two paths to choose from the designated starting point to the designated destination (path A and B, one of which passes through the fire area), the travel time of the two paths is the same. The travel time of the two paths increases because of the traffic congestion

around the fire. In this case, we think the traveler is loss. The travel time of path A is $(t_1, t_2, t_3)$ and the corresponding probability is $(p, q, 1 - p - q)$, the travel time of path B is $(t_4, t_5)$, and the corresponding probability is $(r, 1 - r)$.

In this paper, three kinds of scenes were set in each of the two types of travel modes. The travelers were required to arrive in 15 min while driving, and the travelers were required to arrive in 30 min while walking. Figures 5 and 6 show the specific settings.

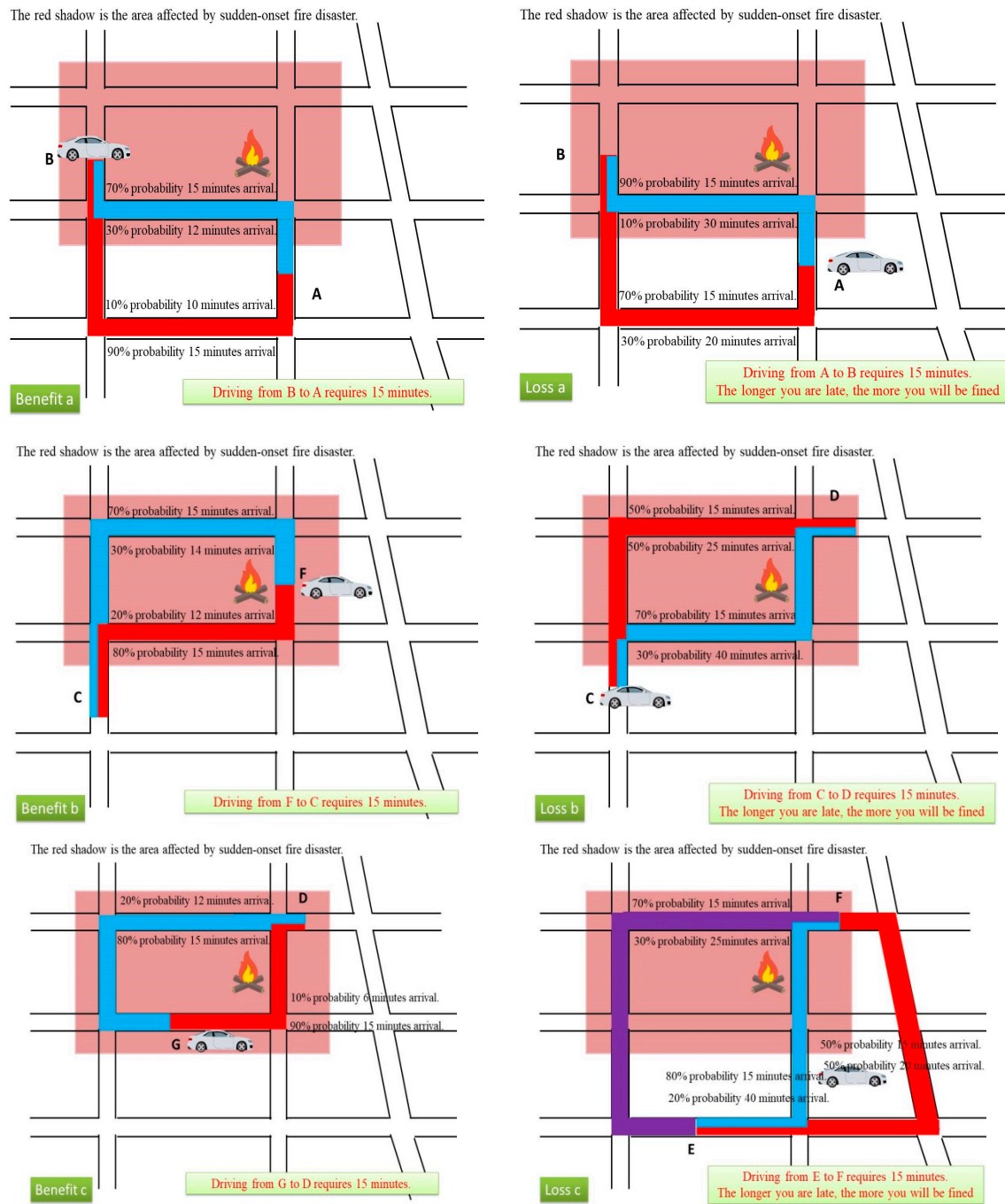

**Figure 5.** Driving through the fire area.

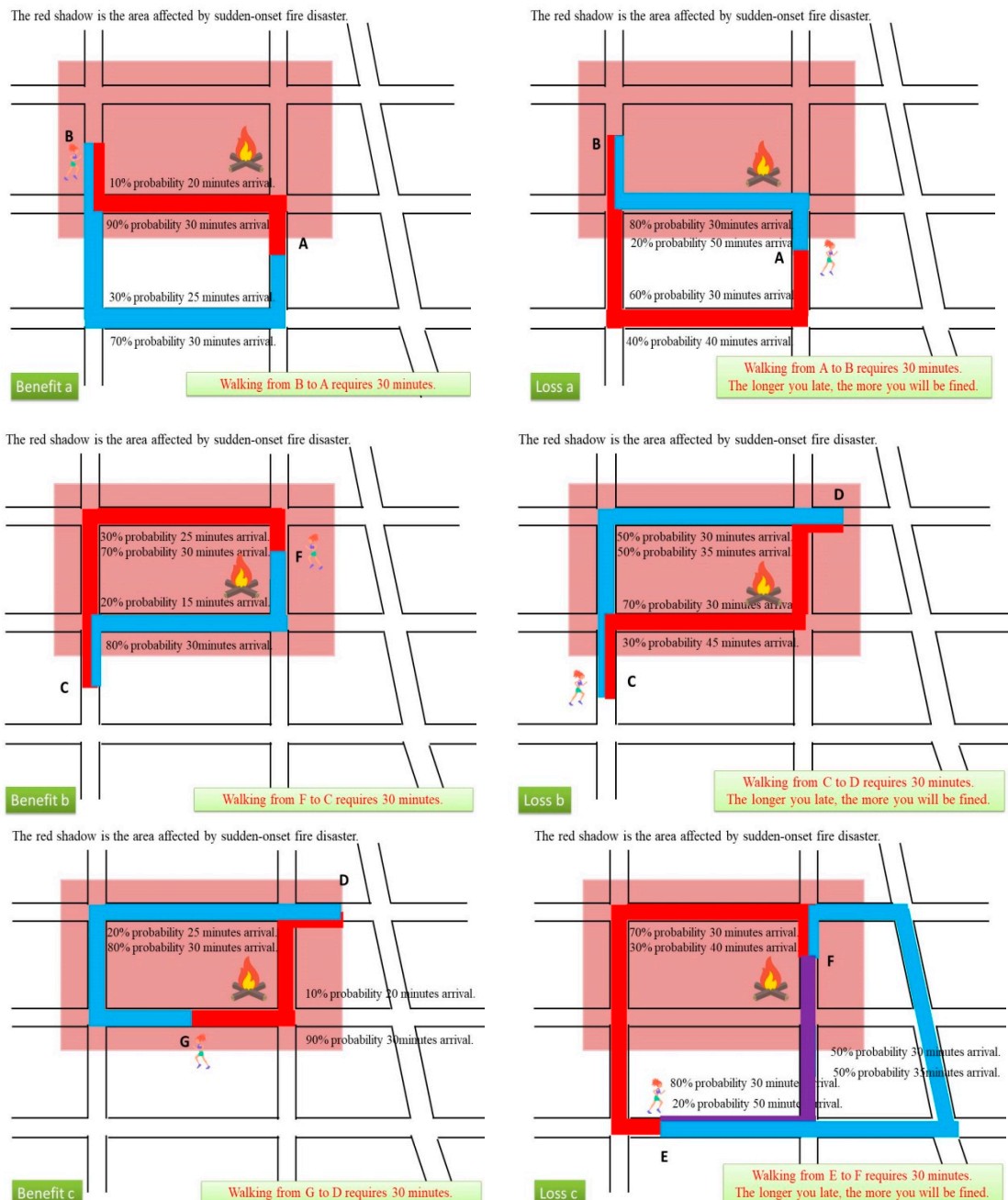

**Figure 6.** Travelers walking through the fire area.

The red shadow part indicates the influence range of fire, as in Figures 5 and 6. In the path setting, the road that travelers pass through the fire area and the road that has not passed through the fire area are set up. The questionnaire set different arrival time and probability through each road. In the benefit scenario setting, traffic control facility near the fire, the passing time is more than the normal time. In the loss scenario setting, the impact of fire on the surrounding roads and traffic congestion occurs, so the time passing through the fire area will be longer.

Table 7 shows the travel time and probability of each path in each scenario. The scenario setting is based on the prospect theory. Take the driving travel benefit and loss scenario 1 as an example. In the driving benefit scenario 1, the traveler will not be late to choose path A or B. The traveler has a 10% probability of 10 min. arrival to choose path A, and 30% probability of 12 min. arrival to choose path B. In this paper, time is used as a measure, the more the traveler chooses path A, the less the probability of early arrival. In the same way, for the loss scenario 1, travelers may be late to choose path A or path

B, 10% of them are late to choose path A, and 30% of them are late to choose path B. However, once path A is late, the time lost is more than path B, and the penalty is more, which is more risky. The setting of other scenes also follows the characteristics of prospect theory.

**Table 7.** Time settings of each scene.

| Travel Mode | Scene | Serial Number | Path A | Path B |
|---|---|---|---|---|
| Drive | Benefit | 1 | (10,0.1;15,0.9) | (12,0.3;15,0.7) |
| | | 2 | (12,0.2;15,0.8) | (14,0.3;15,0.7) |
| | | 3 | (6,0.1;15,0.9) | (12,0.2;15,0.8) |
| | Loss | 1 | (30,0.1;15,0.9) | (20,0.3;15,0.7) |
| | | 2 | (40,0.3;15,0.7) | (25,0.5;15,0.5) |
| | | 3 | (40,0.2;15,0.8) | (20,0.5;15,0.5) |
| Walk | Benefit | 1 | (20,0.1;30,0.9) | (25,0.3;30,0.7) |
| | | 2 | (15,0.2;30,0.8) | (25,0.3;30,0.7) |
| | | 3 | (20,0.1;30,0.9) | (25,0.2;30,0.8) |
| | Loss | 1 | (50,0.2;30,0.8) | (40,0.4;30,0.6) |
| | | 2 | (45,0.3;30,0.7) | (35,0.5;30,0.5) |
| | | 3 | (50,0.2;30,0.8) | (35,0.5;30,0.5) |

*4.2. Analysis of Path Selection Results*

After the questionnaires of the travelers in different scenarios, according to the different risk attitudes, the final path choices of the different risk types of travelers in driving and walking are shown in Tables 8 and 9:

**Table 8.** Traveler's driving route selection results.

| Scene | | Risk Pursuit | | Risk Neutrality | | Risk Aversion | |
|---|---|---|---|---|---|---|---|
| | | Path A | Path B | Path A | Path B | Path A | Path B |
| Benefit | 1 | 29.2% | 70.8% | 32% | 68% | 38.9% | 61.1% |
| | 2 | 36.8% | 63.2% | 39% | 61% | 50% | 50% |
| | 3 | 46.7% | 53.3% | 50% | 50% | 53.3% | 46.7% |
| Loss | 4 | 80.8% | 19.2% | 64% | 36% | 57.8% | 42.2% |
| | 5 | 76.7% | 23.3% | 67% | 33% | 57.8% | 42.2% |
| | 6 | 68.3% | 31.7% | 57% | 43% | 47.8% | 52.2% |

**Table 9.** Traveler walking path selection results.

| Scene | | Risk Pursuit | | Risk Neutrality | | Risk Aversion | |
|---|---|---|---|---|---|---|---|
| | | Path A | Path B | Path A | Path B | Path A | Path B |
| Benefit | 1 | 25.8% | 74.2% | 31% | 69% | 40% | 60% |
| | 2 | 40% | 60% | 41% | 59% | 47.8% | 52.2% |
| | 3 | 46.7% | 53.3% | 47% | 53% | 51.1% | 48.9% |
| Loss | 4 | 82.5% | 17.5% | 71% | 29% | 56.7% | 43.3% |
| | 5 | 71.7% | 28.3% | 62% | 38% | 64.4% | 35.6% |
| | 6 | 59.2% | 40.8% | 53% | 47% | 52.2% | 47.8% |

In the questionnaire setting, 1–3 is the beneficiary area and 4–6 is the loss area. There is a partial probability that the loss area will be late and the time fluctuation of Path A is greater than the Path B. As can be seen from Tables 8 and 9, for all travelers, when they benefit in different scenarios, they tend to be risk-averse, and they tend to pursue risk when they are damaged. This is consistent with the findings of risk attitudes in prospect theory.

### 4.3. Parameter Calibration

In this paper, the parameter values $\gamma$ and $\delta$ are set as $\gamma = \delta = 0.71$, which is the same as Wu and Gonzalez [14]. We used 200 survey data to calibrate the value function of the prospect theory to obtain the different types of risk attitude corresponding parameters. If the traveler is driving, the reference point is set as 15 min. If the traveler is walking, the reference point is set as 30 min. Each path is compared with a reference point to obtain the corresponding benefit and loss values.

Wu [11], Prelec [12], Senbil [13], Stott [27], and others have shown, through experiments, that the parameters in the foreground theory are not fixed and the size of the parameters is affected by the impact of the research object and the scene.

(1) In the questionnaire, in the two scenarios of driving and walking, scenes 1–3 are all beneficial. Therefore, the calibration of the prospect theoretical value function only involves parameters $\alpha$ and $\gamma$, the value of $\gamma$ is 0.71. Hence, we can use two types of scenes 1–3 to calibrate the parameter $\alpha$.

In this survey, the traveler has two optional paths, namely $i = A, B$. When the prospect value of path A is greater than path B, path A is selected, otherwise choice path B. The likelihood function of the sample is as shown in formula 9 while assuming that the survey data are independent of each other:

$$L(\alpha) = \prod_{J=1}^{J} (P_J^i)^{y_J^i}, y_J^i = \begin{cases} 0 \\ 1 \end{cases} \tag{9}$$

where j = 1,2 ... j represents a total of $J$ samples, $P_J^i$ represents the model prediction probability of sample $J$ selection path $i$, as calculated by formula 8. $y_J^i$ indicates whether the sample $J$ selects the path $i$. When sample $J$ does not select path $i$, $y_J^i$ is 0 and when $J$ chooses path $i$, $y_J^i$ is 1. Taking the logarithm of the likelihood function yields formula 10:

$$LogL(\alpha) = \sum_{J=1}^{J} y_j^i LogP_j^i \tag{10}$$

Combined with formula 4, the simplified likelihood function is as shown in formula 11:

$$LogL(\alpha) = \sum_{J=1}^{J} \left( y_j^A Log\left(\frac{\exp(V_j^A)}{(\exp(V_j^A) + \exp(V_j^B))}\right) + y_j^B Log\left(\frac{\exp(V_j^B)}{(\exp(V_j^A) + \exp(V_j^B))}\right) \right) \tag{11}$$

The optimum parameter value is the one that can maximize formula 11. The convergence is poor since the likelihood function of the prospect theory is a nonlinear transcendental equation. Therefore, in this paper we use genetic algorithm to solve the likelihood function and program in MATLAB to solve the genetic algorithm.

(2) In the questionnaire, Scenarios 4–6 are loss. The prospect theoretical value function needs to be calibrated for parameters $\beta$, $\lambda$, and $\delta$, where $\delta = 0.71$, so only need to calibrate $\beta$ and $\lambda$. Similarly, the calibration process is calculated as in formula 12–14:

$$L(\beta, \lambda) = \prod_{j=1}^{J} (P_j^i)^{y_j^i}, y_j^i = \begin{cases} 0 \\ 1 \end{cases} \tag{12}$$

$$LogL(\beta, \lambda) = \sum_{j=1}^{J} y_j^i LogP_j^i \tag{13}$$

$$LogL(\beta, \lambda) = \sum_{j=1}^{J} ((y_j^A Log(\frac{\exp(V_j^A)}{\exp(V_j^A) + \exp(V_j^B)}) + y_j^B Log(\frac{\exp(V_j^B)}{\exp(V_j^A) + \exp(V_j^B)}))) \tag{14}$$

Similarly, a genetic algorithm is used to calibrate the parameters in MATLAB. The optimum parameter is the parameter that reaches the maximum value.

The parameters are set as follows:

(1) Calibration parameter range: $\alpha > 0$, $\beta < 1$, $0 < \lambda < 4$.
(2) Initial population: According to the experience of processing data size, the initial population number is set to 80, which is generated by a random method.
(3) Random operator: select individuals from the group in the way of roulette gambling. The higher the individual fitness is, the higher the probability of being selected. The crossover probability is set to 0.8 and the mutation probability is 0.01.
(4) Termination condition: set the number of iterations to 500.

*4.4. Analysis of Calibration Parameters*

In this paper, the K-T model refers to the results of Kahneman and Tversky's [10] research. The parameters in the traveler's value function of different risk types are calibrated, and Tables 10 and 11 show the risk preference coefficient and the loss avoidance coefficient:

**Table 10.** Traveler driving factor calibration results.

| Parameter | Risk Pursuit | Risk Neutrality | Risk Aversion | K-T Model |
|---|---|---|---|---|
| Risk preference coefficient $\alpha$ | 0.5163 | 0.5982 | 0.5368 | 0.8800 |
| Risk preference coefficient $\beta$ | 0.7932 | 0.8143 | 0.8911 | 0.8800 |
| Loss avoidance factor $\lambda$ | 1.4800 | 1.3200 | 1.1300 | 2.2500 |

**Table 11.** Traveler walking coefficient calibration results.

| Parameter | Risk Pursuit | Risk Neutrality | Risk Aversion | K-T Model |
|---|---|---|---|---|
| Risk preference coefficient $\alpha$ | 0.6302 | 0.6345 | 0.6708 | 0.8800 |
| Risk preference coefficient $\beta$ | 0.8924 | 0.8808 | 0.8941 | 0.8800 |
| Loss avoidance factor $\lambda$ | 2.5100 | 2.3900 | 2.4600 | 2.2500 |

By analyzing Tables 10 and 11, we can get:

(1) Among the two modes of travel, the risk preference coefficient $\beta$ of the three groups of people in the face of loss is greater than the risk preference coefficient $\alpha$ when facing the benefit. In economics, the K-T model research shows that the risk preference coefficient $\beta$ and the loss risk preference coefficient $\alpha$ are equal, which indicates that the research results in economics are not applicable to transportation.

(2) Among the two modes of travel, the risk preference coefficients $\alpha$ and $\beta$ of the three types of risk travelers are less than 1, and the risk preference coefficient at return is smaller than that of the K-T model, and the risk preference coefficient at loss is larger than that of the K-T model, which indicates that travelers are less sensitive to time when they are benefiting than to money, but when they lose, they are more sensitive to their own safety than to money.

(3) The risk appetite coefficient $\alpha$, $\beta$ and the loss avoidance coefficient $\lambda$ of the traveler are greater than the coefficient value when driving, which indicates that traveler driving is more sensitive to the

risk than walking, and the walker will bypass the surrounding areas of sudden disasters as compared with driving.

(4) Among the two modes of travel, when the traveler chooses to walk, the risk preference coefficient $\beta$ and the loss avoidance coefficient $\lambda$ are greater than the coefficient value when driving, which indicates that the traveler is more sensitive to risk when walking.

### 4.5. Models Comparative Analysis

In this paper, while learning from the experimental method of Zhou [28], the remaining 110 data were used to verify the model. We use the study result to compare with the model that was proposed by Kahneman and Tversky [10] and the model proposed by Wang Yan [17]. In the K-T model, the parameter values were set as $\alpha = \beta = 0.88$, $\lambda = 0.25$, $\gamma = 0.69$, and $\delta = 0.61$. In Wang Yan's model, the parameter values were set $\gamma = \delta = 0.71$. In this paper, we used 110 data to verify the model. Table 12 shows Wang Yan's parameter calibration results:

**Table 12.** Wang Yan's parameter calibration results.

| Parameter | Risk Pursuit | Risk Neutrality | Risk Aversion |
|:---:|:---:|:---:|:---:|
| $\alpha$ | 0.43 | 0.34 | 0.26 |
| $\beta$ | 0.45 | 0.49 | 0.41 |
| $\gamma$ | 1.65 | 1.36 | 1.08 |

In this paper, there are two criteria, Mean Absolute Error (MAE) and Mean Absolute Percentage Error (MAPE), which were were used to evaluate the final prediction accuracy. MAE was used to evaluate the prediction bias at the level. MAPE is used to calculate the mean of the absolute differences between the predictive and observed travel choice. Therefore, these two measures were used to evaluate the accuracy and precision of prediction results. The formula of MAE and MAPE are as follows:

$$MAE = \frac{1}{n}\sum_{i}^{n}\left|y_i - \hat{y}_i\right| \tag{15}$$

$$MAPE = \frac{1}{n}\sum_{i=1}^{n}\left|\frac{y_i - \hat{y}_i}{y_i}\right| \times 100\% \tag{16}$$

where $y_i$ is the observed traffic variables, $\hat{y}_i$ is the predicted traffic variables, and $n$ is the total sample size.

The predicted results in this paper take the average of the three types of prediction results. The other two prediction methods are compared with the averages of three path selection types that by survey. The path selection probability is calculated while using the Logit model. Table 13 shows the three prediction results and errors:

**Table 13.** Analysis of the result of selecting path A in different scenarios.

| Way of Travel | Scene | True Value | K-T Model | | | | Wang Yan's Model | | | | This Paper | | | |
|---|---|---|---|---|---|---|---|---|---|---|---|---|---|---|
| | | | Prediction Results | Error | MAE | MAPE | Prediction Results | Error | MAE | MAPE | Prediction Results | Error | MAE | MAPE |
| Drive | 1 | 0.33 | 0.63 | 0.3 | | | 0.52 | 0.19 | | | 0.36 | 0.03 | | |
| | 2 | 0.42 | 0.28 | −0.14 | | | 0.29 | −0.13 | | | 0.4 | −0.02 | | |
| | 3 | 0.5 | 0.72 | 0.22 | 0.21 | 44% | 0.57 | 0.07 | 0.1 | 24% | 0.47 | −0.03 | 0.04 | 7.4% |
| | 4 | 0.68 | 0.53 | −0.15 | | | 0.6 | −0.08 | | | 0.7 | 0.02 | | |
| | 5 | 0.67 | 0.52 | −0.15 | | | 0.61 | −0.06 | | | 0.63 | −0.04 | | |
| | 6 | 0.58 | 0.88 | 0.3 | | | 0.69 | 0.11 | | | 0.67 | 0.09 | | |
| Walk | 1 | 0.32 | 0.53 | 0.21 | | | 0.41 | 0.09 | | | 0.24 | −0.08 | | |
| | 2 | 0.43 | 0.26 | −0.17 | | | 0.52 | 0.09 | | | 0.51 | 0.07 | | |
| | 3 | 0.48 | 0.62 | 0.14 | 0.19 | 38.5% | 0.39 | −0.09 | 0.1 | 19.4% | 0.53 | 0.05 | 0.05 | 8.8% |
| | 4 | 0.7 | 0.92 | 0.22 | | | 0.81 | 0.11 | | | 0.67 | −0.03 | | |
| | 5 | 0.66 | 0.48 | −0.18 | | | 0.78 | 0.12 | | | 0.62 | 0.04 | | |
| | 6 | 0.55 | 0.76 | 0.21 | | | 0.63 | 0.08 | | | 0.5 | 0.05 | | |

Between the two modes of travel, the predicted and true values of the three methods in each scenario are shown in Figures 7 and 8:

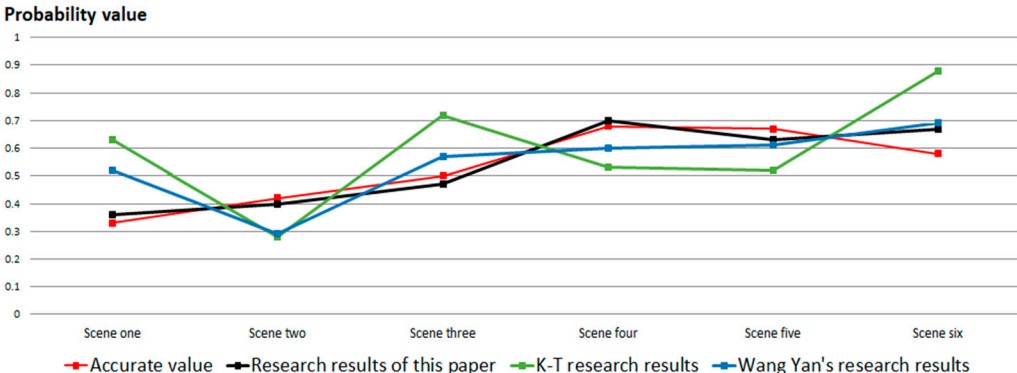

**Figure 7.** Driving error map.

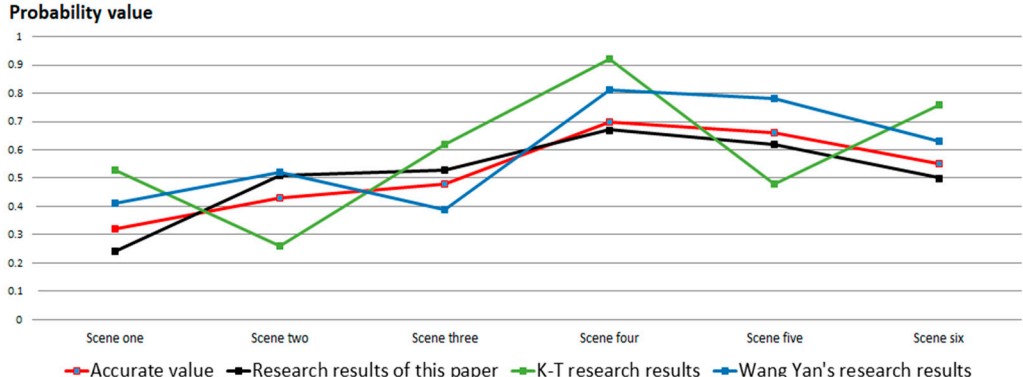

**Figure 8.** Walking error map.

From Figures 7 and 8, the prediction results can be concluded as:

(1) While using the results of K-T model [10], the parameters in economics are applied to the choice of residents' paths, which is the biggest difference from the actual value, no matter for driving or traveling. The error of K-T model is the largest, and the error value and the real value have no stable regularity, because the K-T model is a study on the risk attitude of people in the economic field, which has achieved good research results in the economic field, but, in the traffic travel, especially in the emergency disaster environment, the research conclusions in the use of economy are not consistent with the behavior of the traveler. It shows that the parameters in K-T are only applicable in economics. For traffic scenarios, the research results of K-T are obviously unreasonable; it needs to be analyzed according to specific scenarios.

(2) In the way of driving, the absolute error of the results that was predicted by Wang Yan [17] is between K-T [10] and that predicted by this paper, and the error is approximately 20%. This shows that the traveler's physiological and psychological fluctuation is greater under the specific fire disaster. It is far from predicting the actual path choice of travelers only by studying risk attitude. Therefore, it is necessary to study travelers' risk attitude according to specific scenarios, so as to better analyze travelers' actual choice behavior under sudden disasters.

(3) In the path selection that was predicted by the parameters of this paper, the absolute error of driving and walking is 9% and the average percentage absolute error is below 10%. In the case of sudden fire disaster, the results of this paper have the highest accuracy in the traveler's path selection as compared with the results of other models.

## 5. Conclusions

The research results of this paper aim to solve the emergency traffic problem under the sudden fire disaster in central area of the city. Through the method that was proposed in this paper, people who were walking and driving were analyzed separately in different types of travelers' path selection behaviors that were caused by different physiological psychology in sudden fire disaster. In the questionnaire, travelers were classified and analyzed according to the physiological and psychological factors through setting scenes. Afterwards, we set a travel model that was based on prospect theory in sudden fire disaster. The experimental results show that the maximum prediction error was 9%. The accuracy is higher than other models. It shows that under the sudden fire disaster, analyzing the travel behavior of different types of travelers, has a good effect on the rapid evacuation of traffic in sudden disaster areas and it has good practical significance.

This paper is based on urban sudden fire disaster. The travelers were classified and tried according to the different personal attributes of travelers and different travel modes, and found that good research conclusions were obtained. The road network structure set up in this paper is a simple road network structure and the classification analysis of travelers only considers personal attributes. In the future research, we consider the vehicle hardware in the loop simulation scene to obtain the selection behavior of the travellers in different environments, so as to analyze more accurate factors. At the same time, it is necessary to take the larger road network into account and it is important to analyze the impact of surrounding road conditions for the travelers' path selection.

**Author Contributions:** Conceived and designed the experiments: C.L., D.W. Performed the experiments: B.G., K.W. Analyzed the data: C.L., K.W. Contributed reagents/materials/analysis tools: C.L., K.W., D.W., B.G. Wrote the paper: C.L., K.W., D.W. All authors have read and agreed to the published version of the manuscript.

**Funding:** The authors acknowledge the National Natural Science Foundation of China (Grant No. 51408257 and 51308248), Youth Scientific Research Fund of Jilin (Grant No. 20180520075JH), and the Science and technology project of Jilin Provincial Education Department (Grant No. JJKH20170810KJ and JJKH20180150KJ) are partly support this work.

**Acknowledgments:** We are very thankful to the reviewers for their time and efforts, their comments and suggestions have greatly improved the quality of this paper.

**Conflicts of Interest:** The authors declare no conflict of interest.

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
