# Peer review of "Research on Residents’ Travel Behavior under Sudden Fire Disaster Based on Prospect Theory"

_sustainability, doi:10.3390/su12020487_

Round 1

Reviewer 1 Report

I think this is a very well-written article addressing an important topic. The research design is sound and the method is appropriate. I would like to support this manuscript being accepted and published. 

Author Response

Dear Reviewer:

Manuscript ID: sustainability-630478

Title:“Research on Residents' Travel Behavior under Sudden Fire Disaster Based on Prospect Theory”

We wish to express our very deep appreciation, and the appreciation of all of us, to your great efforts and approval of our manuscript. In the future, we will work harder and more seriously.

We tried our best to improve the manuscript and made some changes in the manuscript. These changes will not influence the content and framework of the paper. We appreciate for your warm work earnestly, and hope that the correction will meet with approval. Thank you for your time and patience. I look forward to receiving your letter.

Yours sincerely,

Authors

Reviewer 2 Report

the paper focuses on the analysis and modelling of travellers' decision-making process with reference to the uncertain/risk contexts of choice. a web-based experiment has been designed and the users' preferences have been collected.

some main doubts are listed below:

the literature review must be updated and integrated also considering some studies about latent factors; a further explanation of the questionnaire (please, provide some snapshot and a more detailed general overview) is required; please, provide further considerations about non-investigated attributes such as attitudes and perceptions (non - observable factors affecting users' behaviour);  it must be clarified in the manuscript the difference between uncertainty and risk as well as the difference between driving and travelling contexts of choices.

Author Response

Dear Reviewer:

Manuscript ID: sustainability-630478

Title:“Research on Residents' Travel Behavior under Sudden Fire Disaster Based on Prospect Theory”

We wish to express our very deep appreciation, and the appreciation of all of us, to your great efforts and suggestions for our manuscript. They are valuable and very helpful for revising and improving our paper, as well as the important guiding to our researches.

The following is a point-to-point response to your comments and responses are in red. The modification marked in red in revised version.

Response to comment:( the literature review must be updated and integrated also considering some studies about latent factors)

Response: We have added references according to the reviewer 's comments .We have added some recent key contributions references and some studies about latent factors. Due to the lack of research on this topic, our newly added reference is the recent research results in line with the topic.

Carlo G P, S.B., Cristina P, Latent variables and route choice behavior. Transportation Research Part B Methodological., 2012: p. 39: 229-319.

Xuan Di, H.X.L., Boundedly rational route choice behavior: A review of models and methodologies. Transportation Research Part B Methodological., 2016: p. 85: 142-179.

Heilman, R.M., Emotion regulation and decision making under risk and uncertainty. Emotion, 2010: p. 10(2): 257-265.

Jou Rongchang, C.K., An application of cumulative prospect theory to freeway drivers’ route choice behaviours. Transportation Research Part A, 2013: p. 49: 123-131.

Response to comment:(a further explanation of the questionnaire (please, provide some snapshot and a more detailed general overview) is required)

Response: We're sorry we didn't explain the details of the questionnaire. We conduct online questionnaire survey through the most authoritative "questionnaire star" Application in China, which can be spread in the form of Wechat group and QQ group, and the application can count the final survey results. In the questionnaire, we have three parts: personal attribute survey, risk attitude classification survey and travel route choice survey. Each kind of survey is explained in detail in the questionnaire. In this paper, the specific investigation content is also described in detail. Some details and statistical results of the questionnaire survey are shown in the following pictures. We conduct communication survey through Wechat group and circle of friends, and there is no time limit to ensure that the respondents have sufficient time to make serious choices. According to the types of travelers in the city, the Wechat group we spread includes the group of ordinary office workers, the group of government employees, the group of teachers and college students, the group of the elderly, etc., which basically includes all the travel groups in the city. From the beginning of the survey, we have made a comprehensive consideration of the respondents to prevent the occurrence of unreasonable data as much as possible, and from the statistical results after processing, the survey data obtained is in line with our expectations, and we think it is basically in line with the actual travel situation. We've added more details based on the reviewers' comments. See line 172-183. The snapshot of questionnaire can be seen in attachment.

Response to comment:(please, provide further considerations about non-investigated attributes such as attitudes and perceptions (non - observable factors affecting users' behaviour))

Response: At the beginning of the survey, we also considered such factors as attitudes and perceptions, but because these factors are hard to quantify and need to be analyzed according to specific scenarios, it is difficult to obtain them, so we ignored these factors and only selected six personal attributes as influencing factors. After fitting the equation, the coefficient and correction coefficient are small, which also shows that the factors not considered will have an impact on the results, so it needs to be further studied in the future. Our paper also explains this, as shown in the line 262 marked red part. In the future research, we consider the vehicle hardware in the loop simulation scene to obtain the selection behavior of the travellers in different environments, so as to analyze more accurate factors. At the same time, according to the reviewer's comments, we add more explanation , See line 225-229.

Response to comment:(it must be clarified in the manuscript the difference between uncertainty and risk as well as the difference between driving and travelling contexts of choices.)

Response: We are sorry that the difference between uncertainty and risk is not clearly described in this paper. In the questionnaire, we have two parts to deal with this problem. In the investigation of risk attitude, we assume that there are uncertain factors in the road conditions, and the risk is that travelers may be late when choosing different roads, and will be fined when they are late. Therefore, the first part is the difference between the uncertainty factors of the road and the risk of fines that travelers may have. In the investigation of the travel behavior in case of sudden fire, the time of the road and arrival probability in each scene, we design it according to the prospect theory. Assuming that the conditions of all the travel roads are basically similar, in each scenario, the road affected by the fire may have the risk of congestion or life-threatening, and there will also be uncertainty factors for smoother traffic. Congestion is due to the chaos of traffic order at the scene of fire. Smooth is due to the police at the scene of fire have made timely traffic control, which is smoother than in normal conditions, and these uncertain factors of benefit or damage just accord with the characteristics of prospect theory. We think the risk is that travelers may face late fines, because it is very important for workers to pay no fines for arriving on time. In our questionnaire, we investigated the drivers of traffic characteristics, ignoring the drivers who traveled through them. According to the reviewer's comments, we add more explanation , See line 184-196.

We tried our best to improve the manuscript and made some changes in the manuscript. These changes will not influence the content and framework of the paper. We appreciate for your warm work earnestly, and hope that the correction will meet with approval. Thank you for your time and patience. I look forward to receiving your letter.

Once again, we would like to thank you for the constructive comments and suggestions. Please feel free to contact us with any questions. We are looking forward to your reply.

Yours sincerely,

Authors

Reviewer 3 Report

This paper addressed a unique issue in travel behaviour analysis. Authors investigated the travel risk attitude from 310 responses though online survey questionnaires and then proposed a prospect theory-based model to analyse the relationship between traveller' personal attributes and risk attitudes. Paper is well structured and written. My main comments are: (1) the justification of the data sufficiency is not given. Also, it is not clear how to avoid data bias through an open data collection method. (2) the literature review needs to be improved as the contribution of the selected topic to existing studies is not clear. (3) the comparative analysis is very supportive to demonstrate the validity. The interpretation of Figures 7 and 8 needs to be more clear. E.g it seems results from K-T model are different from others. 

Author Response

Dear Reviewer:

Manuscript ID: sustainability-630478

Title:“Research on Residents' Travel Behavior under Sudden Fire Disaster Based on Prospect Theory”

We wish to express our very deep appreciation, and the appreciation of all of us, to your great efforts and suggestions for our manuscript. They are valuable and very helpful for revising and improving our paper, as well as the important guiding to our researches.

The following is a point-to-point response to your comments and responses are in red. The modification marked in red in revised version.

Response to comment:(the justification of the data sufficiency is not given. Also, it is not clear how to avoid data bias through an open data collection method.)

Response: We are sorry we didn't explain the source of the data and its validity. We conduct online questionnaire survey through the most authoritative "questionnaire star" Application in China, which can be spread in the form of Wechat group and QQ group, and the application can count the final survey results. Some details and statistical results of the questionnaire survey are shown in the following pictures. We conduct communication survey through Wechat group and circle of friends, and there is no time limit to ensure that the respondents have sufficient time to make serious choices. According to the types of travelers in the city, the Wechat group we spread includes the group of ordinary office workers, the group of government employees, the group of teachers and college students, the group of the elderly, etc., which basically includes all the travel groups in the city. We conducted a three-month investigation. From the beginning of the survey, we have made a comprehensive consideration of the respondents to prevent the occurrence of unreasonable data as much as possible, and from the statistical results after processing, the survey data obtained is in line with our expectations, and we think it is basically in line with the actual travel situation. We've added more details based on the reviewers' comments. The snapshot of questionnaire can be seen in attachment.

Response to comment:(the literature review needs to be improved as the contribution of the selected topic to existing studies is not clear.)

Response: We have added references according to the reviewer 's comments .We have added some recent key contributions references and some studies about latent factors.

Carlo G P, S.B., Cristina P, Latent variables and route choice behavior. Transportation Research Part B Methodological., 2012: p. 39: 229-319.

Xuan  Di, H.X.L., Boundedly rational route choice behavior: A review of models and methodologies. Transportation Research Part B Methodological., 2016: p. 85: 142-179.

Heilman, R.M., Emotion regulation and decision making under risk and uncertainty. Emotion, 2010: p. 10(2): 257-265.

Jou Rongchang, C.K., An application of cumulative prospect theory to freeway drivers’ route choice behaviours. Transportation Research Part A, 2013: p. 49: 123-131.

Response to comment:(the comparative analysis is very supportive to demonstrate the validity. The interpretation of Figures 7 and 8 needs to be more clear. E.g it seems results from K-T model are different from others.)

Response: We are sorry that the comparison results of Figure 7 and figure 8 are not explained clearly. We've added more details based on the reviewers' comments. See line 461.

We tried our best to improve the manuscript and made some changes in the manuscript. These changes will not influence the content and framework of the paper. And here we did not list the changes but marked in red in revised paper. We appreciate for your warm work earnestly, and hope that the correction will meet with approval. Thank you for your time and patience. I look forward to receiving your letter.

Once again, we would like to thank you for the constructive comments and suggestions. Please feel free to contact us with any questions. We are looking forward to your reply.

Yours sincerely,

Authors

Round 2

Reviewer 3 Report

Authors have addressed my comments.